# The Real-World Effects of Route Familiarity on Drivers’ Eye Fixations at Urban Intersections in Changsha, China

**DOI:** 10.3390/ijerph19159529

**Published:** 2022-08-03

**Authors:** Lin Hu, Guangtao Guo, Jing Huang, Xianhui Wu, Kai Chen

**Affiliations:** 1School of Automotive and Mechanical Engineering, Changsha University of Science and Technology, Changsha 410114, China; hulin@csust.edu.cn (L.H.); guangtaoguo@sohu.com (G.G.); 2College of Mechanical and Vehicle Engineering, Hunan University, Changsha 410082, China; 3Key Laboratory of Traffic Safety on Track, Ministry of Education, School of Traffic and Transportation Engineering, Central South University, Changsha 410075, China; wuxianhui9@stu.csust.edu.cn; 4Hunan Judicial Police Vocational College, Changsha 410125, China; 13397529256@163.com

**Keywords:** traffic safety, route familiarity, eye fixation strategy, urban intersections, accident risk

## Abstract

A crucial factor, route familiarity, can affect traffic safety. Nevertheless, focus on the influence of route familiarity on drivers’ eye fixations at urban intersections has received less attention. Identifying the real-world effect of route familiarity on drivers’ eye fixations at urban intersections in Changsha, China, was the objective of this study. Their visual fixation indicators were recorded while unfamiliar drivers and familiar drivers drove a 9 km-long route with nine intersections in an urban environment, but their effectiveness was indicated by the data collected 150 m before the lane stop and 50 m after the lane stop at these intersections. From the analysis of the extracted data, the results indicated that route familiarity could influence drivers’ processing times in the left window (LW) and other areas (OT). Compared with familiar drivers, unfamiliar drivers had longer processing times and higher mental workloads for the right front (RF). For the vehicle’s front (RF, FL, FR), the sampling rates and mental workloads of unfamiliar drivers were higher than those of familiar drivers, but it was the opposite for the driver’s sides (LW, RW) and rear (LM, RM, ReM). It was also indicated that the phenomenon said to increase familiarity with the route and make drivers more likely to be distracted in urban intersections had not been found. From the present findings, the effect of route familiarity on drivers’ eye fixations at urban intersections was confirmed. The high accident risk of familiar drivers could be partly explained by the decrement in drivers’ eye fixation strategies. However, the strategies could not account for the phenomenon that more familiar drivers are involved in rear-end accidents. Therefore, the reason can be investigated based on drivers’ visual scanning strategies, their physiological signals and driving behavior in the future.

## 1. Introduction

In recent years, drivers’ route familiarity has been of great concern to researchers at home and abroad [1]. In related research, there are only two standards to measure a driver’s route familiarity: the O-D distance dimension [2,3] and the driving frequency dimension [4,5,6]. Some studies have pointed out that drivers on familiar routes tend to engage in more dangerous driving behavior, such as proceduralist driving, that can make drivers distracted and more confident in their driving skills [7,8], high speed [9,10,11] and short headway distance [12,13]. This indirectly indicates that road familiarity is a critical cause of road traffic accidents [14,15]. Intini et al. [11] defined familiar and unfamiliar drivers to study the relationship between road familiarity and road accidents according to the drivers’ distance from their residence through a large amount of accident data. The results showed that drivers are more prone to traffic accidents on familiar roads. Some researchers, including Noland [16], also explored the reasons for these conclusions and found that most familiar drivers will attempt to maximize their mobility to reduce their travel time. Therefore, accident risk increases due to more dangerous driving behavior and high driving speeds [15]. Others also believe that driving on familiar roads is more likely to cause distraction and inattention [17] and drivers’ blind confidence in their driving skills [18].

Further, route familiarity affects drivers’ eye fixations. In this aspect, research methods have mainly been driving simulation experiments and on-road tests. Based on the method of driving simulation research, the impact of unfamiliar traffic signs on drivers’ attention allocation was investigated by collecting gaze fixation points and the duration of gaze fixation, and the results indicated that drivers spent more time fixating on unfamiliar signs [19]. Furthermore, drivers could fail to notice changes in traffic signs with increased road familiarity, as was found by Harms and Brookhuis [10]. Based on on-road tests, Babic et al. [20] found that drivers’ perception of traffic signs decreased when they were familiar with the roads and road conditions. Investigating an expert driver, Young et al. [5] also found that dwell time on safety-relevant aspects was affected by route familiarity. However, only Martens and Fox [21] compared the results obtained by an on-road test with the results obtained by a driving simulator, and they found that there were some differences in fixation times and frequencies in relation to specific objects. Based on on-road tests, some also found that the number of fixations and the mean number of fixations, total fixation duration, mean fixation duration and fixation frequencies at some areas of interest (AOIs) are affected by drivers’ familiarity [4,22].

Intersections are among the most important places in a city and therefore are the main focus of researchers [23,24,25]. More than 60% of severe injuries in traffic accidents occur at intersections in European countries [26]. In addition to poor intersection design, road accidents occurring at intersections may also be triggered by drivers’ inappropriate behaviors in multiple aspects of the human–vehicle–road–environment system [27,28,29], as traffic accidents and eye fixation strategies are closely related [30]. The uncertainty of pedestrian crossings and the uncertainty and high speed of nonmotorized vehicles are challenges [31,32] that drivers of motor vehicles may confront at any time. Successfully passing an intersection requires considerable visual information and a high mental load.

Many studies have investigated the effect of drivers’ internal factors, such as age and driving experience, on their attention allocation at intersections [33,34,35]. The study of Anstey, Wood and Lord et al. [36] showed that older drivers’ accident occurrence is primarily due to age-related changes in various aspects of visual attention, including selective attention, divided attention and sustained attention. However, many researchers, such as Hanson and Hildebrand, found that older drivers are involved in fewer accidents at intersections [37]. Although road familiarity is a critical factor in traffic safety, especially at urban intersections, it has not yet attracted considerable research interest.

Although fruitful studies on attention allocation at intersections are already in existence [24,38,39,40], research on the impact of route familiarity on drivers’ eye fixations at intersections is scarce. The objective of this study was to analyze the influence of route familiarity on drivers’ eye fixations at urban intersections using various representations of measures (e.g., duration of fixations, the average duration of fixations and number of fixations at each area of interest (AOI)). The method of identifying the standard of route familiarity is based on driving frequency, and we conducted an on-road experiment in downtown Changsha, China. The total duration of fixations, the average duration of fixations and the number of fixations at each AOI with increased route familiarity were compared to study the drivers’ visual fixation behavior with increasing route familiarity at urban intersections. The contributions of this research include:(a)A summary of the effect of drivers’ familiarity on their visual attention at urban intersections in Changsha, China;(b)Analysis of the mechanism of traffic accidents at intersections;(c)The conclusion that, by taking drivers’ route familiarity into account on roads where intelligent vehicles and ordinary vehicles are mixed, intelligent vehicles’ control decisions can contribute to road traffic safety and enable the anthropomorphic pace of intelligent vehicles to increase.

## 2. Methods

### 2.1. Design

This study mainly controlled driving factors, route familiarity with unfamiliar drivers and familiar drivers. The dependent variables were the total duration of fixations, the average duration of fixations and the number of fixations at each AOI. In addition, frame-by-frame analysis was conducted through Tobii Pro Lab to identify drivers’ eye fixations at urban intersections.

Before the recruitment of participants was executed, the experiment was approved by an ethics committee, Xiangya School of Public Health IRB.

### 2.2. Participants

According to the statistics of the China Traffic Administration Bureau of the Ministry of Public Security on the ratio of male to female drivers in 2016, 18 men and 6 women from universities and local communities were selected for the study. Wu and Xu [41] defined familiar drivers and unfamiliar drivers: drivers that drove the route once a year were regarded as non-familiar drivers, and drivers that drove the route at least once per week were regarded as familiar drivers. Hence, on the basis of their driving frequency over the experimental route at ordinary times before the experiment started, they were preliminarily divided into two distinct groups: an experimental group that included 12 drivers (including 9 men and 3 women) who drove the route at least once a month and a control group that included 12 drivers (including 9 men and 3 women) who drove the route once a year at most, see Table 1 below for relevant information concerning the subjects. Meanwhile, before the experiment was conducted, all participants carefully read and signed informed consent forms. Adequate visual acuity was also ensured. Due to possible interference with the eye movement measurements, none of them wore glasses or hard lenses.

### 2.3. Apparatus

The experimental equipment included an eye-tracking system, the Tobii Pro Glasses 2, and an electric vehicle. As shown in Figure 1, the Tobii Pro Glasses 2 consist of a head-mounted module and storage module, person–vehicle–road–environment–traffic data and eye movement data obtained by the principle that infrared rays are emitted into a driver’s eyes and reflected back to the eye tracking system through his/her pupils. The voice data of these subjects were obtained using a head-mounted module including a scene camera, four eye-tracking cameras and a microphone. These data were superimposed together into a video and a storage module was used to store the video through a data transmission line. Employing the binocular collection method, the system was used to measure the eye movements of 24 participants while driving with a sampling rate of 50 Hz. The drivers wore the eye tracker and drove the same electric vehicle for the on-road test. Although it is safer to use a driving simulator to collect experimental data and allow control of driving conditions, the data obtained by a driving simulator are not as authentic. In addition, the eye movement data measured by the eye tracking system were exported by the dedicated processing software Tobii Pro Lab, and each index datum was obtained by analyzing the video frame by frame.

### 2.4. Experimental Route

Before the experiment, verbal instructions on the study protocol were provided to these participants, especially the preset experimental route and precautions during the experiments. However, they were not informed of the research purpose. The 9 km experimental route, shown in Figure 2 (right), was chosen as an accident black spot [42] described on the basis of accident data from 2014 to 2019 in Changsha city and shown in Figure 2 (left), and consists of 21 intersections among which 9 intersections including unsignalized and signalized intersections were selected, see Table 2. According to specifications for the design of intersections on urban roads and the studies by Intini et al. [14] and Dukic and Broberg [43], the effective distance indicated by the collected data was 150 m before the lane stop line and 50 m after the lane stop line at these intersections, as shown in Figure 3. At the same time, the effective data is the eye movement data of the participants when they pass through the aforementioned intersections.

These participants followed audible GPS instructions on the experimental scenarios for the first time. During the whole experiment, the experimenter sat in the back seat without an additional passenger. To avoid interfering with the quality of the experimental data, the experimenter did not talk to the participants.

### 2.5. Procedure

Both the pre-experiment and the formal experiment began in the middle of July 2020 in good weather without rain and fog. The formal experiment was conducted between 9:00 and 11:00 am or between 14:00 and 18:00 pm, which ensured certain traffic flow. Before either the pre-experiment or the formal experiment, the eye tracking system needed to be calibrated, and only after the system calibration could the experiment be carried out. The pre-experiment proceeded on a non-specified route for the purpose of adapting to the driving process of the experimental vehicle and the eye tracker. For the purpose of making 12 drivers in the control group familiar with the vehicle, the eye tracker was equipped to conduct the pre-experiment for 4 consecutive days, and then the formal experiment was conducted once. Despite this, 12 drivers in the experimental group drove the prescribed route 12 times in 12 consecutive days in order to ensure that they were familiar with the route. During the experiment, distracting objects were not allowed to be carried by the drivers. The data for the 12 drivers in the control group were regarded as those of 12 unfamiliar drivers, and the data on the 12th day for the 12 drivers in the experimental group were considered to be those of 12 familiar drivers, see Table 3. The participants were allowed to drop out at any time if they felt uncomfortable during the experiments.

### 2.6. Data Collection and Reduction

The impact of route familiarity on drivers’ visual fixation behavior at urban intersections was investigated according to 10 AOIs determined by the drivers’ view, including their view from the right front (RF), front-left (FL), front-right (FR), left window (LW), left mirror (LM), right window (RW), right mirror (RM), rearview mirror (ReM), dashboard (DB) and others (OT). Figure 4 show areas of interest (AOIs) divided by the drivers’ view and the corresponding photograph.

The videos that included the drivers’ fixation location to the driving scene through the eye tracker were analyzed and were used to obtain eye movement data. The 10 AOIs were manually drawn frame by frame, and the relevant data and visual attention allocation were then automatically assessed and derived through dedicated software. The drivers’ route familiarity (familiar or unfamiliar drivers) was also recorded.

Due to the number of crashes that previously occurred near most of the intersections, this study primarily investigated drivers’ visual fixation behavior in the green light phase. The effective data distance for analysis was 150 m before the lane stop line to 50 m after the lane stop line at these intersections.

### 2.7. Visual Fixation Measures

The visual fixation measures, total duration of fixations, average duration of fixations and number of fixations at each AOI were employed to assess drivers’ eye fixations at intersections. These visual fixation measures were compared between familiar drivers and unfamiliar drivers through an independent-samples *T*-test for the two-sample heteroskedasticity hypothesis. The statistical significance level was selected to be 0.05.

## 3. Results

Analysis of the data presented in Figure 5 showed that at urban intersections, both the familiar drivers and unfamiliar drivers paid more attention to the right front, namely RF (approximately 82% and 86%, respectively), followed by front-right, namely FR (approximately 5.9% and 5.5%, respectively) and front-left, namely FL (approximately 3.4% and 3.5%, respectively). Compared with the familiar drivers, the unfamiliar drivers generally spent more time looking to the right front, namely RF (t = 3.13, df (degree of freedom) = 19, *p* = 0.005). The differences between the unfamiliar drivers and familiar drivers for the total duration of fixations at the left window, namely LW (t = −2.99, df = 16, *p* = 0.009) and OT (t = 3.85, df = 22, *p* = 0.016) were statistically significant. For unfamiliar drivers, the total duration of fixations at others, namely OT, was 164% longer than that for familiar drivers. However, no statistical significance was found between the unfamiliar drivers and familiar drivers in other AOIs.

Analysis of the data presented in Figure 6 showed that in regard to the average duration of fixations, familiar drivers and unfamiliar drivers still focused more on the road ahead (RF) from the driver’s view (approximately 18.8% and 21.8%). For the average duration of fixations at the right front, namely RF, the unfamiliar drivers fixated significantly longer than the familiar drivers (t = 2.15, df = 21, *p* = 0.042). No statistical significance was found between the familiar drivers and unfamiliar drivers for the average duration of fixations at other AOIs.

Analysis of the data presented in Figure 7 showed that in all AOIs, fixations at drivers’ front (F), including right front (RF), front-left (FL) and front-right (FR), predominated for the familiar drivers (approximately 88.77%) and unfamiliar drivers (approximately 85.09%) at urban intersections, which showed that F provided a large amount of safety-related information required by both the familiar drivers and unfamiliar drivers in order to pass urban intersections safely. The differences between the two groups in the number of fixations at the right front, namely RF, were statistically significant (t = 2.27, df = 21, *p* = 0.034). The familiar drivers checked the left window (LW) 28.5% more frequently than the unfamiliar drivers (t = −2.229, df = 18, *p* = 0.039). At the same time, route familiarity had a significant impact on the number of fixations at the right window, namely RW (t = −3.045, df = 22, *p* = 0.006), and right mirror, namely RM (t = −2.892, df = 19, *p* = 0.009) RM (t = −2.892, df = 22, *p* = 0.008). In addition, for the number of fixations at other AOIs, no significant differences were found between the familiar drivers and unfamiliar drivers.

Table 4 present the four classifications into which the AOIs were divided from the driver’s perspective. For the driver’s front, the unfamiliar drivers had more frequent fixations than those familiar with the route (t = 2.99, df = 22, *p* = 0.006); in addition, the unfamiliar drivers spent more time fixating on the front than the familiar drivers (t = 2.58, df = 22, *p* = 0.017). However, on the driver’s sides, the familiar drivers had more frequent fixations near urban intersections (t = −3.931, df = 22, *p* = 0.0007) and spent more time looking (t = −2.212, df = 22, *p* = 0.038). Similarly, compared to the unfamiliar drivers, the familiar drivers had more frequent fixations at the driver’s rear (t = −3.048, df = 21, *p* = 0.006).

## 4. Discussion

This study investigated the impact of route familiarity on drivers’ eye fixations at urban intersections in Changsha based on on-road experiments. Ten AOIs determined from the driver’s perspectives were assessed. The total duration of fixations, the average duration of fixations and the number of fixations were utilized to assess drivers’ eye fixations at urban intersections. The average duration of fixations and number of fixations were used to study drivers’ visual attention [21,43], and the total duration of fixations was used to explore drivers’ eye fixations [44]. Hurtado and Chiasson [19] pointed out that the total duration of fixations and the average duration of fixations indicate the longer time that drivers need to process information and higher mental workload, respectively. The number of fixations was considered to be the sampling rate [45]. As shown in this research, the drivers familiar with the route checked the right mirror (RM) 42.9% more than the unfamiliar drivers (*p* = 0.009), which indicates that familiar drivers are more likely to have a higher sampling rate in the right mirror (RM). Regarding the average duration of fixations, the unfamiliar drivers had longer durations at the right front (RF) than the familiar drivers (*p* = 0.042), which shows that unfamiliar drivers have more mental workload in processing information for the right front (RF), and in others (OT) and the left window (LW), it was the opposite (*p* = 0.016 and *p* = 0.009, respectively). Table 2 show that drivers with increased familiarity will reduce the sampling rate and processing time for their front, but will increase them for their side, especially the left window, namely LW (*p* = 0.007).

Analysis of the data presented in Table 2 showed that tens of AOIs were grouped into four classifications from the driver’s perspective, namely the vehicle’s configurations, the vehicle’s front, the driver’s sides and the driver’s rear. The relevant discussion is based on the four classifications.

The vehicle’s configurations (dashboard (DB) and others (OT)) can reflect the driver’s condition of distraction while driving. Based on a driving simulator study, Yanko and Spalek [13] demonstrated that familiar drivers are more prone to be distracted on freeway driving routes; however, it was found from the vehicle’s configurations in Table 2 that such a phenomenon does not exist. Therefore, it is difficult to suggest that drivers are prone to be distracted due to the higher situation awareness requirement in urban intersections. 

The vehicle’s front, composed of the right front (RF), front-left (FL) and front-right (FR), contains large amounts of traffic information for both familiar drivers and unfamiliar drivers. The results of the comparison between familiar drivers and unfamiliar drivers are as follows: compared with familiar drivers, unfamiliar drivers had an increased sampling rate (*p* = 0.006), especially in the right front, namely RF (*p* = 0.034), and spent more time obtaining and processing some information (*p* = 0.017), especially from the right front, namely RF (*p* = 0.005). However, in the process of obtaining and processing that information from their right front (RF), the familiar drivers required a lower mental workload than those unfamiliar with the route (*p* = 0.042), which is consistent with the results obtained by Yanko and Spalek [13], stating that increased familiarity with a route will reduce the mental workload required. During situations of lower mental workload, the attentional resource capacity could be decreased [46]. This also explains why familiar drivers are more likely to be involved in traffic accidents than unfamiliar drivers and demonstrates the conclusion that collisions involving familiar drivers are likely due to a lack of attention [47,48].

The left window (LW) and right window (RW) are the sources for drivers to obtain traffic information on the driver’s sides. Compared with unfamiliar drivers, familiar drivers had a significantly higher sampling rate on the driver’s sides (*p* = 0.0007). Familiar drivers had a longer processing time on the driver’s sides than those unfamiliar with the route (*p* = 0.038), especially the left window, namely LW (*p* = 0.009).

In addition, drivers use the left mirror (LM), right mirror (RM) and rearview mirror (ReM) to obtain traffic information from the rear. Despite the fact that familiar drivers fixate more frequently on the rear than unfamiliar drivers (*p* = 0.006), especially at the RM (*p* = 0.009), interestingly, more familiar drivers are involved in rear-end accidents [14]. Meanwhile, as presented in the study, drivers did not tend to be distracted in urban intersections. Therefore, future efforts could be centered on the reason for drivers’ visual scanning strategies, their physiological signals and driving behaviors.

These findings contribute to the development of intelligent vehicles [49]. However, a limitation of the study is that driving experience and intersection type affect drivers’ visual fixations [45,50]. However, in the study, we only discussed drivers’ visual fixation differences influenced by route familiarity at urban intersections in Changsha. The interaction between different driving experiences and route familiarity and between different intersection types and route familiarity in visual fixation behavior were not examined. Consequently, future efforts should be made to reduce the number of accidents at urban intersections annually. In addition, different turning maneuvers also affect drivers’ visual attention [51], but the present study did not take different turning maneuvers into consideration. Relevant research about the effect of route familiarity on visual fixation behavior in the condition of a turning maneuver should be explored. At the same time, drivers’ eye movements are linked to their brain activities, and it was found that mental fatigue could influence the dynamics of saccadic eye movements. Although drivers’ accident mechanism can be analyzed from different perspectives [52,53], it is more important to analyze the mechanism from visual and physiological perspectives. Specifically, in future studies, drivers’ eye movement data should also be used to quantify their brain activities. For example, the question of whether a driver has mental fatigue should be identified. In addition, the impact of the same warning function and different warning methods on driving safety and economy of vehicle should also be focused on in the future, as in [54,55].

## 5. Conclusions

This study mainly explores the impact of route familiarity on drivers’ eye fixation behaviors at urban intersections. We found that route familiarity differed in drivers’ eye fixation behaviors at urban intersections in different AOIs. In addition, based on the established relationship between eye fixation behavior and the driver’s mental workload, it was found that route familiarity also affects the drivers’ sampling rate, processing time of traffic information and their mental workload. Additionally, the high accident risk of familiar drivers could be partly explained by the decrement in drivers’ eye fixation strategies. These findings dig deeper into why drivers are involved in accidents, and incorporating driver’s route familiarity into AV research can help anthropomorphize AVs.

## Figures and Tables

**Figure 1 ijerph-19-09529-f001:**
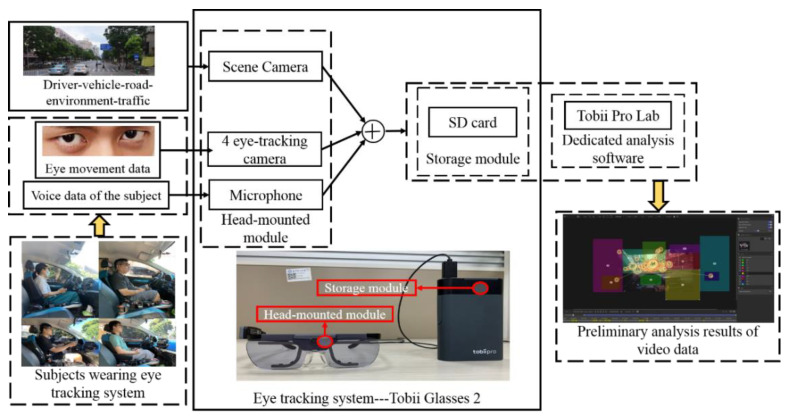
Introduction to the principle of the Tobii Pro Glasses and preprocessing of video data.

**Figure 2 ijerph-19-09529-f002:**
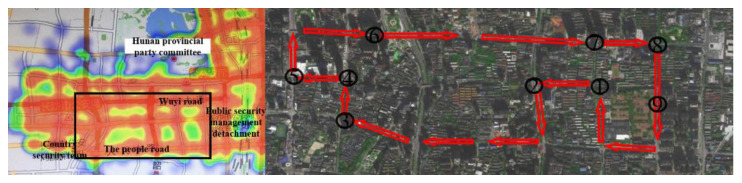
The distribution of traffic accidents in Changsha city from 2014 to 2016 and the selected test route. On the left, orange expresses the largest number of occurred accidents, and the experimental route is selected in the rectangle. On the right, the composition of red arrows indicates the experimental route, and the number denotes urban intersections passed through.

**Figure 3 ijerph-19-09529-f003:**
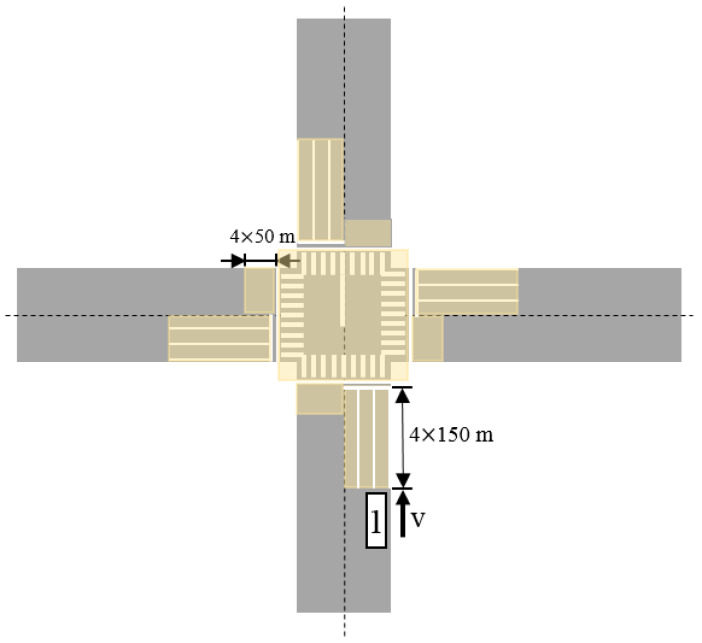
Illustration of the data indicating effective distance at the intersection. The beige area that covers the intersection denotes the effective distance for obtaining data.

**Figure 4 ijerph-19-09529-f004:**
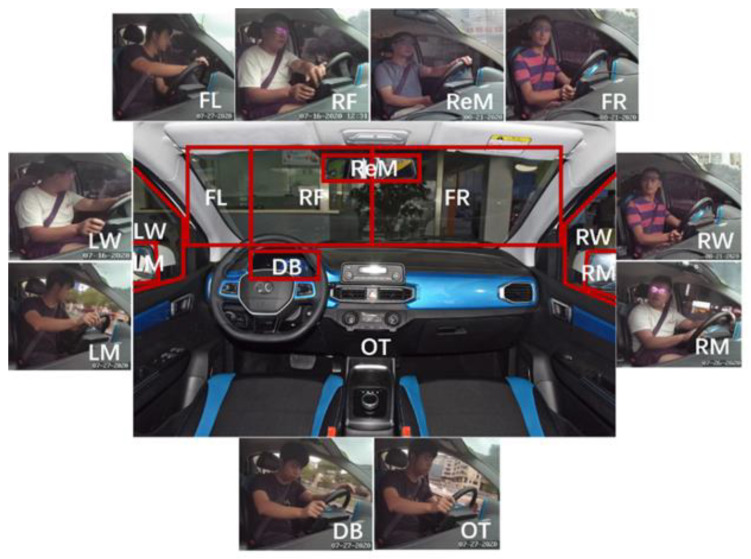
The specific locations of AOIs determined by the drivers’ views and the corresponding photographs. RF refers to right front, FL refers to front-left, FR refers to front-right, LW refers to left window, LM refers to left mirror, RW refers to right window, RM refers to right mirror, ReM refers to rearview mirror, DB refers to dashboard and OT refers to others.

**Figure 5 ijerph-19-09529-f005:**
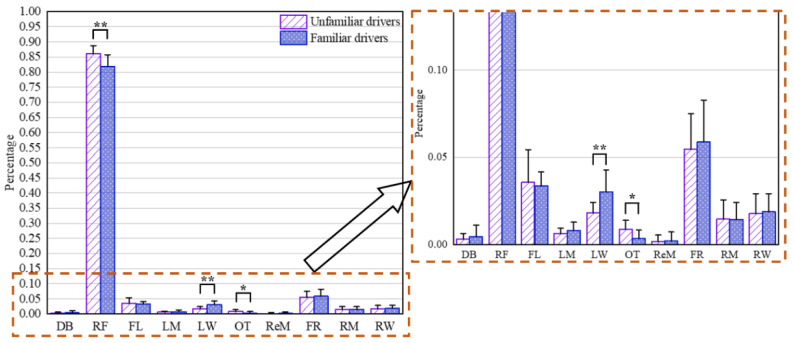
Percentages of total duration of fixations at each AOI at urban intersections for unfamiliar drivers and familiar drivers. The error bar denotes standard deviation. * denotes significance at the 0.05 level (*p* ≤ 0.05). ** denotes significance at the 0.01 level (*p* ≤ 0.01). RF refers to right front, FL refers to front-left, FR refers to front-right, LW refers to left window, LM refers to left mirror, RW refers to right window, RM refers to right mirror, ReM refers to rearview mirror, DB refers to dashboard and OT refers to others.

**Figure 6 ijerph-19-09529-f006:**
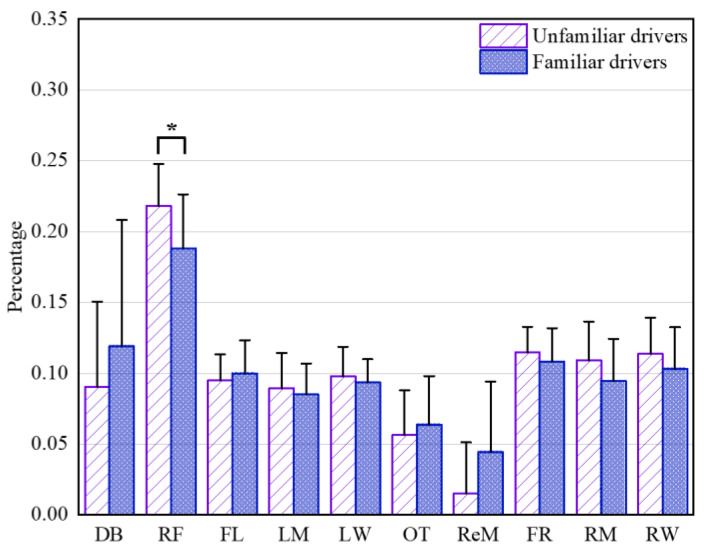
Percentages of average duration of fixations at each AOI at urban intersections for unfamiliar drivers and familiar drivers. * denotes significance at the 0.05 level (*p* ≤ 0.05). RF refers to right front, FL refers to front-left, FR refers to front-right, LW refers to left window, LM refers to left mirror, RW refers to right window, RM refers to right mirror, ReM refers to rearview mirror, DB refers to dashboard and OT refers to others.

**Figure 7 ijerph-19-09529-f007:**
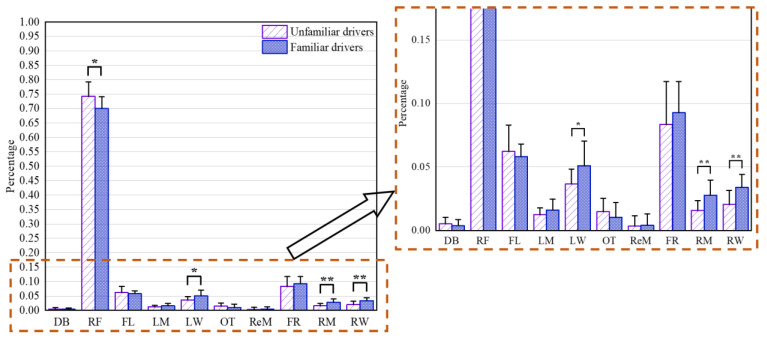
Percentages of the number of fixations at each AOI at urban intersections for unfamiliar drivers and familiar drivers. * denotes significance at the 0.05 level (*p* ≤ 0.05). ** denotes significance at the 0.01 level (*p* ≤ 0.01). RF refers to right front, FL refers to front-left, FR refers to front-right, LW refers to left window, LM refers to left mirror, RW refers to right window, RM refers to right mirror, ReM refers to rearview mirror, DB refers to dashboard and OT refers to others.

**Table 1 ijerph-19-09529-t001:** The relevant information of the subjects.

Type of Drivers	Age (Mean ± SD)	Driving Experience (Mean ± SD)	Number	Driving License	Preliminary Road Familiarity
Experimental group	31.12 ± 7.60	5.52 ± 2.80	12 (9, 3)	√	at least once a month
Control group			12 (9, 3)	√	once a year at most

**Table 2 ijerph-19-09529-t002:** The number of intersections and the number of selected intersections in the experimental route.

Passing Maneuvers	Intersections	Selected Intersections
Left turn	3	3
Right turn	7	3
Straight across	11	3

**Table 3 ijerph-19-09529-t003:** The number of experiments and the selection of data for analysis in this study.

Type of Drivers	Preliminary Road Familiarity	Number of Experiments	Data Selected
experimental group	at least once a month	12	12th experiment data
control group	once a year at most	1	1st experiment data

**Table 4 ijerph-19-09529-t004:** Four classifications into which AOIs were divided from the drivers’ perspective and the comparison between unfamiliar drivers and familiar drivers for each index.

Area Classification of Driving Scenario	AOIs	TDF	ADF	NF
The vehicle’s configurations	DB	-	-	-
	OT			
The vehicle’s front	RF	F < U **	-	F < U **
	FL			
	FR			
The driver’s sides	LW	F > U *	-	F > U **
	RW			
The driver’s rear	LM	-	-	F > U **
	RM			
	ReM			

Notes: TDF refers to the total duration of fixations. ADF refers to the average duration of fixations. NF refers to the number of fixations. F and U represent familiar drivers and unfamiliar drivers, respectively. * and ** denote significant correlations in which the index in the AOI would be affected by route familiarity. * represents *p* < 0.05. ** represents *p* < 0.01.—represents no significant differences between familiar drivers and unfamiliar drivers. RF refers to right front, FL refers to front-left, FR refers to front-right, LW refers to left window, LM refers to left mirror, RW refers to right window, RM refers to right mirror, ReM refers to rearview mirror, DB refers to dashboard and OT refers to others.

## Data Availability

Not applicable.

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
