# Peer review of "The Real-World Effects of Route Familiarity on Drivers’ Eye Fixations at Urban Intersections in Changsha, China"

_ijerph, 2022, doi:10.3390/ijerph19159529_

Round 1

Reviewer 1 Report

All the authors' answers are clear in the presented resubmitted version, the last chapter, the conclusion for which I asked, has also been added.

Author Response

Thank you for your valuable comments!

Please find the attached authors' reply documents.

Reviewer 2 Report

The authors corrected some of the review comments and omitted the rest. The authors have still not adhered to the authors' guidelines. Oddly enough, they even left Section 0 "How to use this template" in the corrected article. So they did not even read that they were to delete this paragraph. In the literature list, instead of the author's name and initials, they write Author 1, Author 2.

Detailed review comments.

  1. Line 42-43. The definition of “unfamiliar drivers and familiar drivers” must be used in the article earlier, i.e. in the place where the authors used this term for the first time, i.e. line 42-43. The explanation to the reviewer is a separate matter, because the reviewer found it in line 170-179 and suggested it to the authors.
  2. Line 75-76. Sentence: To successfully pass an intersection requires considerable visual information and a high mental load.
    Please write an explanation clearly and directly in the article, because it proves that the authors contribute something new, i.e. increase the value of the research and the article.
  3. Subsection 2.2 Participants. There is Table 1 in subsection 2. There is no chart. There are no references to Table and Figures in the text.
  4. References. Correct according to the pattern given in https://www.mdpi.com/journal/ijerph/instructions and in the WORD file ijerph-template.dot.

Author Response

(The authors gave the same response as above.)

Reviewer 3 Report

Dear Authors,

Thank you for your clear responses to my comments.  After corrections, the paper is very interesting and the methodology and results are described clearly.

Regards.

Author Response

(The authors gave the same response as above.)

Reviewer 4 Report

The reviewer thinks the paper can be accepted for publication. However, the reviewer still suggests a language polish.

Author Response

Thank you for your valuable comments!

Please find the attached authors' reply documents.

This manuscript is a resubmission of an earlier submission. The following is a list of the peer review reports and author responses from that submission.

Round 1

Reviewer 1 Report

Dear Authors, 

Thank you for your research and an interesting article about the results. 

One must-see thing missing from your article is the lack of an ending at the end... The "Discussion" chapter is quite long, but the reader does not know your final thoughts about your research. Please add the "Conclusions" section. 

A few questions about your research: 

1) Did the participants know exactly where the experimental path was? 

2) How were people divided into two groups (the main group and the control group)? 

3) (Fig.4) Why does "RF" mean a significance of 0.01 (p≤0.01)"?

Thank you

Author Response

Dear Mr. Ang Seng Chuan,

   I'm very glad to receive your letter about the manuscript. In regard to the 27% replication ratio, I would like to make the following explanation, as follows:

  1. I used the journal template, the replication ratio includes the reproduction ratio that are not part of the body of the paper, see replication ratio for details. And the replication ratio includes the reproduction ratio about references.
  2. The replication ratio includes the reproduction ratio about proper nouns, such as route familiarity, the total duration of fixations, average duration of fixations, number of fixations and so on.
  3. In addition, all others have been modified.  

Reviewer 2 Report

I am very pleased to read this manuscript. However, it requires a few fixes and additions.

Detailed comments

1.      Line 42-43. In the article, please define the concept of unfamiliar drivers and familiar drivers. Does this mean drivers selected for the study and random drivers appearing at the analysed intersection or are drivers knowing a given intersection and drivers appearing at a given intersection by accident and driving intelligent vehicles. It might be best to write that information earlier as the authors only provided it later in subsection 2.5. Procedure. Maybe use the description contained in the lines 170-179.

2.      Line 45-46. Some researchers […] you should list the researches you write about e.g.: Some researchers Intini, Colonna, Berloco and et al. [10] have also explored the reasons and have found ….

3.      Line 75-76. Sentence: To successfully pass an intersection requires considerable visual information and a high mental load. Is this the conclusion of the authors from the analysed literature, i.e. it is necessary to emphasize it and write more about this research gap and define it well? This research gap is the authors' contribution to new research.

4.      Line 82-84. Sentence: Although road familiarity is a critical factor in traffic safety, especially at urban intersections, it has not yet attracted considerable research interest. Similarly as above, write about the identified research gap and about the fact that the authors did just that.

5.      Line 85-86. Sentence: Although fruitful studies on attention allocation at intersections are already in existence, research on the impact of route familiarity on drivers' eye fixations at intersections is scarce.

Please list these studies […]. There is also research by Babkov, Hobbs and Richardson in the literature. And a drawing from these books showing the fixation of the driver's eyesight. Please add it to the article and provide also related articles writing about driver's eyesight fixation.

Hobbs, F.D., Richardson, B.D., Traffic engineering, WKŁ, Warsaw 1971.

Babkov, V.F., Road conditions and traffic safety, WKŁ, Warsaw 1975, Moscow: Mir Publishers, ©1975.

6.      Line 94-100. Please correct and describe it stylistically.

7.      In introduction: This article requires some general organizational work. Using the IMRAD schema can help by making your quotation more organic and can improve your sources organization. It can be more synthetic and better explained in many parts. The introduction requires a huge amount of work as well as methodology.

8.      The purpose of the work as well as the assumptions are not clear.

9.      First part must be reorganized. Mention any limitations of the study. In the Introduction please emphasize the literature gap about this topic that existed before the time this research is published.

10.  Subsection 2.2 Participants. Please supplement the given data with charts. Mark known and unknown drivers on them. Maybe use the description contained in lines 170-179.

11.  Subsection 2.3 Apparatus. Please provide a photo of the Tobii Pro Glasses 2 camera and provide a detailed description of the sensors. Add topart of Figure 3 inside the apparatus and give here the acronyms used by the authors. Explain how pulses are recorded, i.e. what exactly measure the eye movements record, and give an example of video image analysis frame by frame.

12.  Line 150-152. Please correct and describe it stylistically.

13.  Figure 1. Enlarge each part of Figures 1 and put them in two lines one below the other. Give a legend to your drawing. Omit the street names in both parts of Figure 1. Each figure should be at least 350 dpi.

(https://www.mdpi.com/journal/ijerph/instructions:

Preparing Figures, Schemes and Tables

File for Figures and Schemes must be provided during submission in a single zip archive and at a sufficiently high resolution (minimum 1000 pixels width/height, or a resolution of 300 dpi or higher). Common formats are accepted, however, TIFF, JPEG, EPS and PDF are preferred.)

14.  Table 1. Please use the same formatting of text regarding small or Capital letters.

15.  Figure 2. Put a space before the unit

16.  Subsection 2.6. Data collection and reduction. Line 182. How many AOI were there, 10 or 9?

17.  Figure 3. FL give in the top row on the left. RW give in the top row on the right. add a line below them and put DB in the middle here. Under the view of the inside of the cabin, give the OT centre in the centre. Notes regarding dpi as in Figure 1.

18.  Subsection 2.7 Visual fixation measures. Please provide the measurement results in the table with the division into known and unknown drivers. (familiar drivers and unfamiliar drivers).

19.  Line 207-216. Acronyms are hardly legible for the Reader. Please write their full name in front of all the acronyms used here.

20.  Figure 4. For a better understanding of the data given in the graph, please provide the interior of the car from Figure 3 below the graph with the markings, as indicated by the acronym used.

21.  Line 207, 221, 233. In scientific articles please don’t write “As illustrated in Fig.…”, instead please write in scientific language “The analysis of the data presented in Figure… showed that…”

22.  Figure 5. Notes as in Figure 4.

23.  Figure 5. Notes as in Figure 4.

24.  Line 282. Notes as in item 21.

25.  Due to the considerable dispersion of the analysed fixations, it is better to omit acronyms in the text of the article. And use them only on the axes of the graphs with the drawing of the centre added in from Figure 3.

26.  References. Correct according to the pattern given in https://www.mdpi.com/journal/ijerph/instructions and in the WORD file ijerph-template.dot.

Author Response

 I'm very glad to receive your letter about the manuscript. In regard to the 27% replication ratio, I would like to make the following explanation, as follows:

  1. I used the journal template, the replication ratio includes the reproduction ratio that are not part of the body of the paper, as shown in figure 1. And the replication ratio includes the reproduction ratio about references.
  2. The replication ratio includes the reproduction ratio about proper nouns, such as route familiarity, the total duration of fixations, average duration of fixations, number of fixations and so on.
  3. In addition, all others have been modified.

Reviewer 3 Report

The paper is interesting and fits well with the topic of the IJERPH special issue.

Below are some comments to the authors:

 1. The abstract needs improvement. In accordance with the journal's editorial requirements the abstract should be a single paragraph.

2. The terms "familiar drivers" and "unfamiliar drivers" require further explanation.

3. The methodology presented in the paper needs to be clarified.

- Selected intersections are shown in Figure 1. Are the results presented in Section 3 from all intersections or just one of them?

- In the introduction, the authors wrote that the duration of the drivers' eye fixations in the different area was measured. It is unclear whether this refers to the duration of eye fixations in a particular area at one time, or the total duration of eye fixations in a particular area during the entire crossing of the intersection.

- In Figure 2, the beige color indicates the area where the recording was conducted. It is not clear whether the data recording took place while driving through the intersection or also during a stop due to a traffic light change or a traffic jam.

4. A conclusion section in accordance with the journal's editorial requirements is mandatory, so the paper should include this section.

Author Response

(The authors gave the same response as above.)

Reviewer 4 Report

The current study explored the difference of eye fixation between familiar and unfamiliar drivers. This interesting topic can be viewed as an evaluation of driving risk, which is highly related to the road safety. The experiment design seems ok, yet I still have some concerns about the paper. Please see the comments below.

I. The eye movement is typically used in the experiments regarding distracted driving, fatigue driving, etc. since these maneuvers are practically associated with crash risk. Therefore, the readers may want to see more pieces of evidence that how eye fixation could affect the crashes. The relative contents could be enhanced in the Introduction section.

II. The experiment was only conducted with 24 drivers. However, the ideal number of participants in eye movement experiments is 30 or more (see doi.org/10.1016/j.aap.2018.08.024 and doi.org/10.1016/j.aap.2021.106143). This consequently may not be enough to gain robust conclusions by analyzing with statistical approaches. The authors are supposed to comment on it.

III. The authors used independent-samples T-tests in the current study. This approach assumes that the samples are collected from a fixed population so the groups share a same variance. To keep the reliability of the results, the study is supposed to examine the heteroscedasticity between the sample groups.

IV. The percentages of total/average duration of eye fixations at each AOI are compared in the analysis. However, the study didn’t link these characteristics with the driving safety. The authors could examine the effect of them on driving safety using surrogate safety measures such as driving speed, speed variance, conflicts or hard-brake. If this is not allowed in the dataset, the limitations should be stated carefully.

V. Many confusing sentences and phrases make the paper hard to read. The English writing should be polished thoroughly. For example,

1) Use “vehicle front” rather than “vehicle forward

2) In Section 2.6, what does the sentence “On account of occurred crashes near most of the intersection, a vehicle is moving” mean?

3) It is hard to understand the contents from the line 108 to line 112.

4) Line 265 to line 266. “Ten AOIs divided from the driver’s perspectives were assessed”. I think only the features of eye fixation were assessed here rather than the AOIs. Also, you can say “Ten AOIs were defined through xxx.

Author Response

(The authors gave the same response as above.)

Round 2

Reviewer 1 Report

Unfortunately, I am forced to reject the article, I have not received any answers to any of the questions I asked. 
I am not Dear Mr. Ang Seng Chuan.

Reviewer 2 Report

The authors corrected only a very small part of the comments and skipped the rest of them. Taking into account the lack of significant amendments, the reviewer recommends serious amendments to the article before its submission.

1.      The authors didn’t adopted to many authors’ guidelines.

2.      Oddly enough, they even left Section 0 "How to use this template" in the corrected article. So they did not even read that they were to delete this paragraph.

3.      The reviewer did not receive a single answer to the questions asked.

4.      Did the authors receive Reviewer 2's review at all?

5.      There are also no corrections in the corrected article, according to comments from other Reviewers.

6.      As an answer to Reviewer 1, only the private letter of authors to the Assistant Editor Mr. Ang Seng Chuan

Reviewer 4 Report

All the reviewer comments are not carefully addressed. Further, there is no any response to the reviewer comments.

The manuscript still needs a lot of work before any publication.